# Verification Learning:
# Make Unsupervised Neuro-Symbolic System Feasible

**Lin-Han Jia** [1]  **Wen-Chao Hu** [1 2]  **Jie-Jing Shao** [1 2]  **Lan-Zhe Guo** [1 3]  **Yu-Feng Li** [1 2]

## Abstract

The current Neuro-Symbolic (NeSy) Learning paradigm suffers from an over-reliance on labeled data, so if we completely disregard labels, it leads to less symbol information, a larger solution space, and more shortcuts-issues that current Nesy systems cannot resolve. This paper introduces a novel learning paradigm, Verification Learning (VL), which addresses this challenge by transforming the label-based reasoning process in Nesy into a label-free verification process. VL achieves excellent learning results solely by relying on unlabeled data and a function that verifies whether the current predictions conform to the rules. We formalize this problem as a Constraint Optimization Problem (COP) and propose a Dynamic Combinatorial Sorting (DCS) algorithm that accelerates the solution by reducing verification attempts, effectively lowering computational costs and introduce a prior alignment method to address potential shortcuts. Our theoretical analysis points out which tasks in Nesy systems can be completed without labels and explains why rules can replace infinite labels for some tasks, while for others the rules have no effect. We validate the proposed framework through several fully unsupervised tasks including addition, sort, match, and chess, each showing significant performance and efficiency improvements.

## 1. Introduction

Human cognition operates through a dual-system framework: System 1 (intuitive processing) enables rapid,

[1]National Key Laboratory for Novel Software Technology, Nanjing University, Nanjing, China [2]School of Artificial Intelligence, Nanjing University, Nanjing, China [3]School of Intelligence Science and Technology, Nanjing University, Nanjing, China. Correspondence to: Yu-Feng Li <liyf@lamda.nju.edu.cn>, Lan-Zhe Guo <guolz@lamda.nju.edu.cn>.

*Proceedings of the 42$^{nd}$ International Conference on Machine Learning*, Vancouver, Canada. PMLR 267, 2025. Copyright 2025 by the author(s).

experience-based responses to simple tasks, while System 2 (deliberative processing) engages slow, knowledge-intensive reasoning for complex ones. This cognitive architecture mirrors the dichotomy in artificial intelligence (AI) between data-driven and rule-driven learning paradigms. In the past, rule-driven symbolic learning systems were less favored compared to data-driven neural learning systems due to their high maintenance and search costs. However, with the continuous iteration of neural network models, they still perform poorly on reasoning tasks. Consequently, an increasing number of research efforts are now focused on Nesy systems capable of integrating data-driven and rule-driven methods.

In the Nesy paradigm, a machine learning model $f$ establishes a mapping between inputs $X$ and symbolic representations $S$, i.e., $S = f(X)$. Then, using a knowledge base $KB$ and $S$, to infer the label $Y$, i.e., $KB, S \models Y$. $S$ is unknown and needs to be predicted based on both $X$ and $Y$, ensuring consistency between the learning and reasoning processes. While the goal is to leverage knowledge to reduce reliance on labeled data, the real-world application of Nesy remains limited. For simple tasks, good results can be achieved without a symbolic system. For complex tasks, the required data grows exponentially but most current Nesy algorithms still require labels $Y$ of the same scale as the input data $X$ which are expensive to acquire. Therefore, there is an urgent need to develop unsupervised Nesy systems.

The difficulty in bridging the gap between supervised Nesy and unsupervised Nesy remains significant which can be seen in Figure 1. Firstly, the unsupervised paradigm lacks critical information compared to the supervised paradigm. Much of the performance improvement seen in current Nesy algorithms comes from label leakage (Chang et al., 2020). For example, in the common task of handwritten digit addition (Manhaeve et al., 2018), providing labels $Y$ for the equations ($0 + 0 = 0$) and ($9 + 9 = 18$) effectively leaks the symbolic labels 0 and 9, further propagated to other labels, essentially supervising the learning of $X$ directly. Secondly, the lack of labels drastically increases the search space for problem-solving. For example, transforming the problem ($9 + 9 = 18$) to ($9 + 9 = 1, 8$) changes the search space from $10^2$ to $10^4$ and when labels are available, one can further restrict the search to solutions

where $y = 18$. Finally, the absence of labels leads to a significant increase in the shortcuts problem in Nesy tasks. For example, in the equation ($\boxed{9} + \boxed{8} = 17$), the only shortcut is $(8 + 9 = 17)$, while for ($\boxed{9} + \boxed{8} = \boxed{1} + \boxed{7}$), every other equation that satisfies the addition rule like $(0 + 0 = 00)$ forms a shortcut.

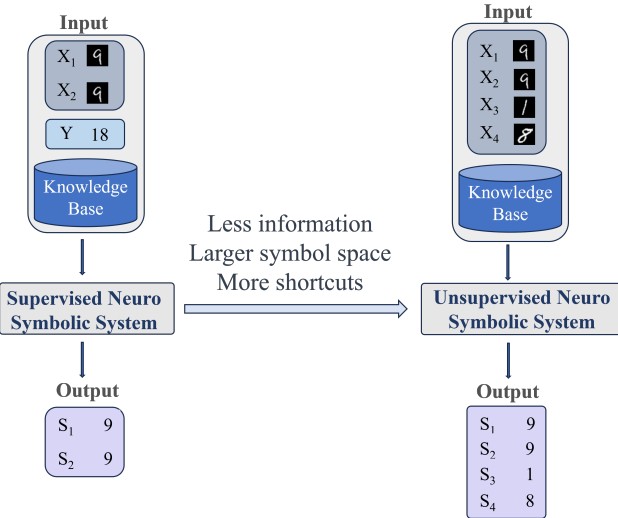

*Figure 1.* This example illustrates the differences between unsupervised and supervised neuro-symbolic systems in the addition task. It highlights that unsupervised neuro-symbolic systems exhibit broader applicability but also pose greater learning challenges.

Due to the aforementioned issues, applying previous approaches for unsupervised Nesy is largely infeasible. To address these problems, we propose a new paradigm called Verification Learning (VL). In the VL paradigm, the model's entire learning process (illustrated in Figure 2) can be completed using only unlabeled data and a knowledge-based verification function. In the unsupervised setting, where a knowledge module cannot perform reasoning starting from the label $Y$, we instead replace the reasoning process with a constraint satisfaction verification on $S$ drawn from the solution space. Using satisfiability verification instead of logical reasoning not only resolves the problem of a lack of starting points, but also bypasses challenges inherent in logical reasoning, such as incompleteness, infinite recursion, and high computational complexity. Additionally, this approach is not tied to a specific logical form, allowing more general verification functions to replace the complex logic programming. Moreover, because it does not depend on label $Y$, the VL paradigm supports test time corrections, ensuring that the solutions output by the model conform to the specified constraints.

In the problem-solving process, our goal is to identify the symbolic labels that maximize the alignment between the knowledge base and the machine learning predictions. VL's learning process corresponds to a Constraint Optimization Problem (COP), where the objective is to find the solution that maximizes the optimization score among all feasible solutions that satisfy the constraints. If the verification of solutions requires exhaustively traversing the entire solution space, this becomes computationally expensive. However, if we can sort the solutions in the solution space according to their scores and verify them in order dynamically, we can ensure that the first valid solution we verify is the optimal one. Initially, this sorting process had exponential complexity, but we introduced an algorithm called Dynamic Combinatorial Sorting (DCS), which is an extension of (Jia et al., 2025). This algorithm maintains a heap structure with low computational overhead, dynamically tracking the solution with the highest optimization goal value among unverified solutions. By doing so, we guarantee that the first valid solution we verify is the optimal one, thus reducing the COP problem's complexity to a similar level as the CSP. Initially, we implemented DCS under the assumption of independence and later extended it to handle some cases where independence does not hold but monotonicity holds, making it still applicable. This reveals that, in Nesy, monotonicity is a more relaxed yet crucial property compared to independence.

Additionally, we propose a distribution alignment method to mitigate the severe shortcut and collapse problems caused by the excessive number of feasible solutions in the unsupervised setting. By leveraging the natural distribution of symbolic systems, this algorithm adjusts the output distribution of the machine learning model, providing the necessary self-correction capability during training.

We also established an effective theoretical framework for unsupervised Nesy, proving that if the model's output symbolic distribution aligns with the natural distribution, the worst-case performance of unsupervised Nesy depends on the number of single-point orbits after performing the symmetry group decomposition of the rule base. The average performance is determined by the size of each orbit corresponding to symbols. This theory also reveals that if two symbols are not completely equivalent in their role within the knowledge base, they can be distinguished under sufficient unlabeled data and optimization ultimately.

We conducted experiments on four rule-based tasks without labels, and made groundbreaking progress. We were able to: identify the numbers in addition expressions based solely on the addition rule (Manhaeve et al., 2018); recognize numbers in ordered sequences based solely on the sort rule (Winters et al., 2022); identify characters in strings based solely on the string match rule (Dai et al., 2019); and identify chess pieces on a chessboard based solely on the chess rule. The experimental results demonstrate the outstanding performance and efficiency of the VL paradigm.

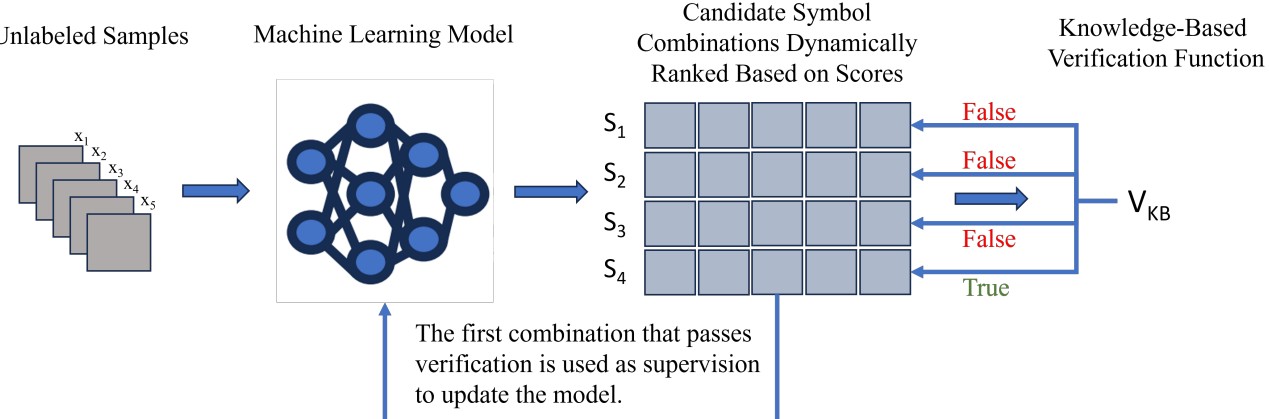

*Figure 2.* This example illustrates how verification learning can complete the entire model training process using only unlabeled data and a knowledge-based verification function.

## 2. Related Works

### 2.1. Neuro-Symbolic Learning

Research on Nesy can generally be classified into three types (Yu et al., 2023): Reasoning to learning, which focuses on constructing network architectures (Mao et al., 2019) or loss functions based on rules, emphasizing the mapping from $X$ to $S$; Learning to reasoning, where learning methods are employed to identify reasoning paths (Mao et al., 2019) or accelerate the search process (Wang et al., 2018), focusing on the reasoning from $S$ to $Y$; Learning-reasoning: which emphasizes the interaction between the two components and where $S$ is unknown and needs to be obtained based on both $X$ and $Y$, ensuring consistency between the learning and reasoning processes (Xu et al., 2018; Stammer et al., 2023; Petersen et al., 2021). Deep-Problog introduces probabilistic reasoning by incorporating probabilities predicted by neural networks into Prolog programs (Manhaeve et al., 2018), while DeepStochlog, based on DeepProlog, uses Stochastic Definite Clause Grammars for stochastic reasoning (Winters et al., 2022). NeurASP, on the other hand, uses Answer Set Programming as the knowledge base (Yang et al., 2023). Among the above methods, all tend toward probabilistic reasoning. In contrast, abductive learning (ABL), as an important branch, is dedicated to inferring a definite symbol $S$ from $Y$, ensuring that $S$ aligns with the neural network's predictions (Dai et al., 2019; Huang et al., 2021; 2020; He et al., 2024; Shao et al., 2025; Jia et al., 2025; Hu et al., 2025b). Ground ABL preprocesses a Ground Truth knowledge base based on ABL to avoid using Prolog search (Cai et al., 2021), while A3BL computes loss based on confidence-weighted calculations for different candidate symbols $S$ (He et al., 2024).

The concept of label leakage was first introduced in the context of SATNet (Wang et al., 2019) solving visual Su-

doku problems (Chang et al., 2020), where the model could directly determine symbolic labels $S$ without relying on the input $X$, instead using only the labels $Y$. After eliminating this leakage, SATNet's performance on digit recognition completely failed, even dropping to 0%. We've found that this phenomenon is widespread in Nesy tasks. This reveals a situation in many tasks where the answer can be obtained without learning, significantly lowering the task's difficulty and failing to accurately reflect the system's ability to integrate the two components.

For the optimization problem of Nesy solving, it often involves search problems with exponential complexity. Currently, there are some works that optimize this, such as using gradient-free optimization algorithms (Yu et al., 2016), predicting error locations (Morishita et al., 2023), and designing reward-based search methods using reinforcement learning algorithms (Hu et al., 2025a; Li et al., 2020). However, these optimization algorithms cannot guarantee that the solution found is globally optimal.

Additionally, Nesy faces the issue of shortcuts (Yang et al., 2024; He et al., 2024). Many studies have highlighted that knowledge bases may have multiple satisfying solutions, but fail to accurately pinpoint the target one. However, few works have provided viable solutions to address this problem.

Recently, van Krieken et al. (2024) explores the issue of independence in Nesy, pointing out that current Nesy frameworks rely on the assumption of independence. However, in reality, symbols do not satisfy independence always.

### 2.2. Constraint Optimization Problem (COP)

A CSP is defined as a triplet (Variable, Domain, Constraint), where "Variable" refers to a set of decision variables, each with possible values from the set "Domain," and "Con-

straint" is the rules that must be satisfied by these variables. A COP is represented as (Variable, Domain, Constraint, Objective), and it seeks to find a feasible solution that optimizes an objective function defined over the variables (Fioretto et al., 2018). In a minimization problem, a solution $S \in COP(Variable, Domain, Constraint, Objective)$ if and only if $S \in CSP(Variable, Domain, Constraint)$, and $\forall T \in CSP(Variable, Domain, Constraint)$, $Objective(S) \leq Objective(T)$ (Mulamba et al., 2020).

## 3. From Supervised Reasoning to Unsupervised Verification

In the classical Nesy paradigm, the perception module contains a learning function that establishes a mapping between input $X$ and symbols $S$. Here, $S$ is a symbolic sequence of length $l$, i.e., $S = [s_1, \ldots, s_l]$, where $s_i \in \mathcal{C} = \{c_1, \ldots, c_k\}$ and $s_i \sim P$, with $k$ representing the size of the symbol space and $P$ representing the natural distribution of the symbols. The entire symbolic sequence space is represented as $\mathcal{S} = \mathcal{C}^L$. The input $X$ may correspond to a sequence of inputs $[x_1, \ldots, x_l]$ that match the meaning of the symbols in $S$, where $x_i \in \mathbb{R}^d$, and $d$ is the dimensionality of $x_i$. The global input space is $\mathcal{X} = \mathbb{R}^{dL}$.

The perception module $f(S|X) \in \mathcal{F}$ maps inputs to the symbol space. Here, $f$ corresponds to the conditional distribution of the intermediate concepts given the input, where $f$ is in the hypothesis space $\mathcal{F} \subset \mathcal{X} \rightarrow \mathcal{S}$. Additionally, we define $g(S|X)$ to represent the model's predicted probability distribution in the symbol space.

The reasoning module contains a knowledge base $KB$. From the input symbolic sequence $S$, we can infer a label $Y \in \mathcal{Y}$, i.e., $S, KB \models Y$. Furthermore, using $Y$ and $KB$, a set of possible candidate solutions for $S$ can also be inferred inversely, i.e., $KB, Y \models candidates(S) = \{S_1, \ldots, S_{|candidates(S)|}\}$, where the ground truth $S$ is guaranteed to be in $candidates(S)$. For any $S_i \in candidates(S) \setminus \{S\}$, it is a shortcut to $S$.

In the training process of a Nesy system, the input dataset $X_{train} = [(X_1, Y_1), \ldots, (X_n, Y_n)]$ learns to maintain consistency between input and output by learning intermediate symbolic sequences $S$. We denote the loss function used as $L$, Methods like DeepProblog, DeepStochlog, and similar approaches minimize the following:

$$\min_f \sum_{S' \in \text{candidates}(S)} \text{Score}(S') \cdot L(f(X), S') \quad (1)$$

Alternatively, approaches like ABL minimize:

$$\min_{f \in \mathcal{F}} L(f(X), S_{opt})$$
$$s.t. \quad S_{opt} = \arg \max_{S' \in candidates(S)} Score(S') \quad (2)$$

Here, $score(S')$ represents the probability or weight associated with $S'$, and the definition of the score varies across different approaches. In DeepProblog and DeepStochlog, the score is the product of the rule probabilities along the reasoning path. In ABL, the score corresponds to the consistency distance between $f(X)$ and $S'$, while in A3BL, it reflects the confidence in $g(X)$.

In real-world scenarios, $Y$ generally does not exist and is more often part of the unknown symbolic sequence $S$. Therefore, in unsupervised Nesy, we must avoid relying on $Y$ as a starting point for reasoning. Instead, $candidates(S)$ is directly determined by the knowledge base $KB$, i.e., $KB \models candidates(S)$. This introduces several challenges: 1. It prevents label leakage of $Y$ into $S$; 2. It significantly increases the space of $S$; 3. More importantly, it leads to an overabundance of $candidates(S)$ because of extreme shortcuts, making the process of traversing and scoring all candidate solutions extremely difficult.

In unsupervised settings, the original reasoning process $KB \models candidates(S)$ and the subsequent traversal of $candidates(S)$ to compute scores and select the best solution or a weighted sum become computationally prohibitive. In practical applications, this makes the system no longer deployable. Therefore, we convert the paradigm of $KB \models candidates(S)$ into a process of generating new solutions $S'$ and validating whether the current predicted symbolic sequence $S'$ belongs to $candidates(S)$. Specifically, if $S', KB \models True$, then we can prove that $S' \in candidates(S)$.

This verification paradigm is simpler and more aligned with real-world scenarios for several reasons: 1. It is suitable for unsupervised settings and does not require a known starting point $Y$ for reasoning, which also enables verification to support test time corrections when reasoning cannot; 2. Verification does not rely on a complete knowledge base. According to the incompleteness theorem, most real-world symbolic systems (even simple arithmetic systems) are incomplete, meaning there are many true propositions without provable reasoning paths but can be validated as true; 3. Verification guarantees the process is halting, whereas reasoning cannot guarantee this due to recursions; 4. The computational complexity of validation is typically much lower than that of reasoning. Many NP problems can be validated in polynomial time, but finding a solution cannot be done yet within polynomial time; 5. Verification is generally more convenient and general at the programming level. It only requires a function $V_{KB} : \mathcal{S} \rightarrow \{True, False\}$, which can replace the entire knowledge base. It is not limited to propositional logic or first-order logic and does not require constructing a search process like in Prolog.

With this approach, we bypass the complex processes involved with $candidates(S)$. Moreover, due to the exces-

sive shortcuts in unsupervised settings, methods like Deep-Problog, which find all candidate solutions and weight them, are no longer applicable. Instead, we adopt a strategy similar to ABL, where we select only the highest-scoring solutions. Our goal now shifts to finding the highest-scoring solution that can be verified, i.e., solving the COP problem $COP(S, \mathcal{S}, V_{KB}, Score)$. In the following, we will explore how to solve the COP problem with minimal cost.

## 4. Solve COP by Dynamic Combinatorial Sorting

Solving the optimization problem $COP(S, \mathcal{S}, V_{KB}, Score)$ requires us to find the solution that satisfies the $V_{KB}$ constraint and has the highest score. Following conventional methods, we would need to verify all possible assignments in the feasible set to determine which solution satisfies the constraint and has the highest score. This exponential time complexity approach is not feasible. Therefore, we propose a strategy that validates solutions in descending order of score in $\mathcal{S}$. By doing so, we ensure that the first solution validated by $V_{KB}$ successfully is the optimal one.

Given that $S = [s_1, \ldots, s_l]$ consists of $l$ symbols, and considering the global feasible set as a combination of multiple variables, this problem is known as the combinatorial sorting problem. Since the size of the feasible set is $k^l$, using classical sorting algorithms would be computationally infeasible. However, when the score satisfies certain properties, We can infer the $(i + 1)_{th}$ ranked solution based on the top $i$ known solutions. In this way, we only need to know the highest-scoring solution to sequentially derive all other solutions, without having to enumerate and sort all of them. This significantly reduces the computational complexity while ensuring that the verification order remains strict, guaranteeing the global optimality of the COP solution.

First, we discuss the scenario where the score satisfies independence (van Krieken et al., 2024), and present a combinatorial sorting solution. We then extend the solution to cases where independence is not satisfied but monotonicity is, which provides more relaxed conditions. This highlights that monotonicity is a more crucial property for Nesy tasks, and problems that satisfy monotonicity are just as easy to solve as those that satisfy independence.

### 4.1. Independent Case

In the case where the score has independence, $Score(S) = \prod_{s_i \in S} score(s_i)$, where $score(s_i)$ represents the score of an individual symbol in the symbol combination $S$, such as the commonly used confidence $g(X)_{i,s_i}$. Under the independence assumption, we sort all possible values for each symbol $s_i$. This ensures that for each position, the assignment with the highest score is selected. When a symbol $s_{i,j}$

is changed to $s_{i,j+1}$, the score for that position $score(s_{i,j})$ will change to $score(s_{i,j+1})$, which updates the global score $Score(S)$ by a factor of $v_i = \frac{score(s_{i,j+1})}{score(s_{i,j})}$, where $v_i$ represents the minimal cost of changing symbol $s_i$. For the current unverified assignment $S$, we select the next assignment by sorting the successors based on the smallest cost $v_i$ and iteratively update the assignments. Hence, each assignment $S$ has at most $l$ successors, and the successors are ordered. We denote these successors as $Suc(S)$.

Under the property of independence, while the greedy rule does not hold (i.e., the $(i + 1)_{th}$ assignment $S_{i+1}$ is not necessarily the successor of the $i_{th}$ assignment $S_i$), it is always true that $S_{i+1}$ is a successor of one of the preceding $i$ assignments. Formally, for any $i \in k^l$, there exists $1 \leq j \leq i$ such that $S_{i+1} \in Suc(S_j)$.

**Theorem 4.1.** *When the Score satisfies independence, i.e., $Score(S) = \prod_{s_p \in S} score(s_p)$, for any $i > 1$, there exists $0 < j < i, 0 < p \leq l, 0 < q < k$, such that modifying $s_p \in S_j$ from the original symbol $s_{p,q}$ to $s_{p,q+1}$ results in $s_{p'}$, and $Suc(S_j)_p = (s_1, \ldots, s_{p-1}, s_{p'}, s_{p+1}, \ldots, s_l) \in Suc(S_j)$. Furthermore, there does not exist any $S' \in \mathcal{S}$ such that $S' \notin \{S_1, \ldots, S_{i-1}\}$ and $Score(S') > Score(Suc(S_j)_p)$. Therefore, $S_i = Suc(S_j)_p$.*

This crucial property allows us to maintain a heap structure dynamically, where we track the successors of each verified assignment, keyed by the maximum successor score. This enables us to quickly find the assignment with the highest score among all unverified successors of the first $i$ assignments, ensuring that the assignment ranked $i + 1$ is selected.

Next, the following process can be used to dynamically find the next assignment to be validated:

1. For $i = 1$, we select the highest-scoring assignment for each position in $S_1$, and derive the successors $Suc(S_1)$, which are then placed in the heap.

2. For $i > 1$, we select the highest-scoring successor from the heap, i.e., the one with the maximum successor score $S_j$, and then select the highest-scoring assignment from $Suc(S_j)$ to form $S_i$. We then derive the successors $Suc(S_i)$, which are then placed in the heap. The successor assignments of $S_j$ are updated to $Suc(S_j) \setminus \{S_i\}$, and if non-empty, they are placed back in the heap.

By following this process, we ensure that the assignments are verified in strict score order, guaranteeing that the first solution to pass the verification is the optimal solution for the COP problem. If the first valid solution is ranked $K$, the complexity of maintaining the heap structure is $O(K \log K)$, and the total complexity becomes $O(K(\log K + l \log l + k \log k))$.

## 4.2. Non-Independent Case

The selection of the Score function can vary, and affects the performance of VL. Using more diverse optimization objectives often leads to violations of the independence assumption. For instance, the consistency score $Score(S) = \sum_{i=1}^{l}[S_i = f(X)_i]$ in (Dai et al., 2019) distance cannot be expressed as an independent score.

We examined whether DCS remains applicable in broader contexts. We found that the fundamental reason for DCS success lies in Theorem 4.1: the $i + 1$-th assignment comes from one of the previous $i$ assignments by modifying only one symbol. If this condition is violated, it means that at least one position has had a change in priority between different symbols, thus affecting the global score $Score(S)$. As a sufficient condition, we can guarantee that if the priority between different symbols at the same position satisfies a fixed total ordering, DCS will always find the optimal solution to the COP problem.

**Theorem 4.2.** *When the Score satisfies monotonicity, i.e., for any $0 < p \le l$ and any $0 < q < k$, there does not exist a $S'$ such that modifying $s_p$ from $s_{p,q}$ to $s_{p,q+1}$ results in $s_{p'}$, and $Suc(S')_p = (s_1, \ldots, s_{p-1}, s_{p'}, s_{p+1}, \ldots, s_l) \in Suc(S')$, satisfying $Score(Suc(S')_p) > Score(S')$, then for any $i > 1$, there exists $0 < j < i$, $0 < p \le l$ results in $Suc(S_j)_p = (s_1, \ldots, s_{p-1}, s_{p'}, s_{p+1}, \ldots, s_l) \in Suc(S_j)$. Furthermore, there does not exist any $S' \in \mathcal{S}$ such that $S' \notin \{S_1, \ldots, S_{i-1}\}$ and $Score(S') > Score(Suc(S_j)_p)$. Therefore, $S_i = Suc(S_j)_p$.*

**Proposition 4.3.** *Clearly, the satisfaction of independence by Score is a sufficient but not necessary condition for Score to satisfy monotonicity.*

For the consistency distance, symbol $s_i$ matches the predicted value $f(X)_i$ at position $i$ is given higher priority, while the priorities for other symbols remain the same. This satisfies the monotonicity property. Therefore, even with consistency score, combinatorial sorting can still be applied successfully. Furthermore, if we combine multiple Score functions that satisfy monotonicity (e.g., first prioritize the assignments with the highest consistency, then select the ones with the highest confidence), DSC can still achieve the optimal solution to the COP problem.

It is worth noting that when monotonicity is not satisfied, even obtaining the top-ranked solution requires exhaustively searching the entire solution space with exponential complexity. Therefore, the following theorem holds.

**Theorem 4.4.** *The monotonicity of the score function is a sufficient and necessary condition for the COP problem to admit a general algorithm with sub-exponential complexity.*

If even monotonicity is violated, solving for the optimal $Score(S)$ without traversing all solutions becomes impos-

sible. In such cases, strategies like reinforcement learning or other sampling-based methods can be used to estimate $Score(S)$. However, these methods no longer guarantee a strict order, and thus, cannot ensure the optimal solution to the COP problem.

# 5. Mitigate Shortcut Problem by Distribution Alignment

After setting up the framework, we can begin learning using unsupervised data and the verification procedure. However, there are significant challenges in unsupervised scenarios, particularly due to the shortcut problem. Additionally, during the initialization phase of the neural network, the predictions are often highly biased, leading to a situation where the network explores only a few symbols. For example, in an addition task, if all symbols are predicted as 0 (i.e., $0 + 0 = 00$), it can pass the verification and also minimize the loss in the learning task to zero in the addition task.

To address this, we propose a distribution alignment strategy. By aligning the model's output symbol distribution with the natural distribution $P$ of the symbols, we can significantly mitigate this issue. In cases where the natural distribution is unknown, we can use a uniform distribution as a prior. During the initial phase, this ensures that the output distribution is spread out. In later stages, even after removing the influence of the prior distribution, the model's output distribution will eventually concentrate on the distribution consistent with the training data. The specific adjustment process is to directly modify the probabilities output by the model to:

$$g(X)_{i,j} = \frac{l \cdot P_{s_j} \cdot g(X)_{i,j}}{\sum_{k=1}^{l} g(X)_{k,j}}. \tag{3}$$

# 6. Theoretical Study on When Knowledge Can Assist Unsupervised Learning

We performed a theoretical analysis of VL which relies solely on unlabeled data and a verification function. The theoretical results demonstrate the upper bounds that Nesy can achieve under the condition of distribution alignment, without requiring supervision. We further analyzed which tasks can be solved without labels and which cannot.

In fact, the theoretical analysis of unsupervised learning can be divided into two parts. The first component is the knowledge-induced error, which reflects the system's ability to establish a correspondence between the categories identified by the learner and the labels of the symbols. The second part is the data-induced error, which reflects the system's ability to group samples belonging to the same category together. Together, these two factors determine the upper bound of the generalization error in unsupervised learning.

VL effectively addresses both of these issues, with the two modules working together to promote each other's success.

## 6.1. Knowledge-Induced Error in VL

VL can make the predicted results align with the rule base. So our primary goal is to explore the gap between maintaining consistency with the rule base and being able to distinguish between symbols. This gap is determined by the symmetry of the function $V_{KB}$ on the symbol set.

Let's start with a symbol set $S = \{s_1, \ldots, s_k\}$, and define its symmetry group, $Sym(S)$, which consists of all possible permutations of the set. Any permutation $\sigma \in Sym(S)$ is a bijection $\sigma : \mathcal{S} \to \mathcal{S}$. We define $G$ as the symmetry group of the verification function $V_{KB}$, i.e., the subgroup of $Sym(S)$ that keeps the results of $V_{KB}$ invariant. Specifically, for all $S \in \mathcal{S}$ and $\sigma \in G$, let $\sigma(S) = (\sigma(s_i))_{s_i \in S}$ we have $V_{KB}(S) = V_{KB}(\sigma(S))$.

We perform an orbit decomposition of the symmetry group $G$ for the function $V_{KB}$. For each symbol $s_i$, its orbit is defined as $O_{s_i} = \{\sigma(s_i) | \sigma \in G\}$, which represents all the equivalence symbols that can be reached by applying the symmetries of $G$ to $s_i$.

If the orbit $O_{s_i}$ of symbol $s_i$ is a singleton (i.e., $O_{s_i} = \{s_i\}$), then $s_i$ is a fixed point of $V_{KB}$. Therefore, by the orbit decomposition, we can identify the set of fixed points of the verification function $V_{KB}$ as $Fix(G) = \{s_i | O_{s_i} = \{s_i\}\}$. The task-induced upper bound on error is given by the sum of probabilities of the symbols that are not fixed points, i.e.,

$$R_{task}^{up} = \sum_{s_i \in \mathcal{S}} \mathbb{I}(s_i \notin Fix(G)) P_{s_i} \qquad (4)$$

The average error is determined by the sum of the ratio of the symbol probabilities to the sizes of their orbits:

$$R_{task}^{avg} = \sum_{s_i \in \mathcal{S}} \frac{P_{s_i}}{|O_{s_i}|} \qquad (5)$$

This error, caused by the task itself, cannot be compensated by data, making it a fundamental limitation that cannot be addressed through knowledge verification. To better understand these concepts, consider the following examples:

1. Sudoku Task: In the Sudoku task, any permutation of numbers still satisfies the constraints. This means that all permutations of symbols in Sudoku do not alter the knowledge base. Consequently, the upper bound on task-induced error for Sudoku is $R_{sudoku}^{up} = 100\%$, meaning that regardless of data size or model performance, the fully unsupervised Sudoku task could still have a 0% accuracy.

2. Addition Task: In the addition task, no permutation of symbols (such as numbers) can still satisfy the addition

system unless the symbols correspond to the correct values. Thus, all symbols in the addition task are fixed points, and the upper bound error is $R_{addition}^{up} = 0\%$. This means that with enough data and model improvements, we can achieve good results.

3. Chess Task: It is also important to consider cases where a one-way inclusion (not a bijection) occurs. For example, in a chessboard scenario, a Queen can move along diagonal and straight lines, a Rook along straight lines, and a Bishop along diagonals. This leads to situations where a Rook or Bishop could be mapped as a Queen. In such cases, the task becomes unsolvable because it lacks the necessary bijections. However, the presence of a natural distribution $P$ helps correct this issue by ensuring that, the model can distinguish Rooks and Bishops from Queens. So $R_{chess}^{up} = 0\%$. Thus, the distribution $P$ extends the domain of solvable problems, enabling the model to handle more complex tasks that would otherwise be unsolvable.

## 6.2. Data-Induced Error in VL

Once the error induced by the task is determined, combining it with the error introduced by the learning process allows us to obtain the upper bound on the generalization error. In unsupervised learning, since it is not possible to distinguish between the symbols in the same orbit, the empirical error should be minimized by selecting the permutation in the symmetry group $G$. We define $\hat{R}(f)$ as the current minimal symmetric permutation empirical error.

$$\hat{R}(f) = \min_{\sigma \in G} \sum_{X \in X_{\text{train}}} [g(X)_i \neq \sigma(s_i)] \qquad (6)$$

By minimizing the permutation error and the task-induced error, we can derive the performance bounds for VL in solving unsupervised neural-symbolic learning tasks.

**Theorem 6.1.** *For any function $f \in \mathcal{F}$, if $L$ is a $\rho$-Lipschitz continuous loss function, $\mathcal{R}_n$ is the Rademacher complexity for a sample size of $n$, $R_{task}^{up}$ is the current task-induced upper bound on error, and $\hat{R}(f)$ is the current minimal symmetric permutation empirical error, then the empirical error $R(f)$ for the prediction of the current symbol set by $f$ satisfies, with at least probability $1 - \delta$:*

$$R(f) \leq \hat{R}(f) + 2\rho\mathcal{R}_n(F) + 3\sqrt{\frac{log(2/\delta)}{2n}} + R_{task}^{up} \quad (7)$$

## 7. Experiments

To validate the effectiveness of the framework we proposed, we conducted experiments on 4 unsupervised tasks. These experiments were primarily extensions of previously supervised Nesy tasks. For all tasks, we used LeNet as the basic

*Table 1.* The comparison of symbol recognition accuracy on the Addition dataset.

| Method | 2 | 3 | 4 | 5 | 6 | 7 | 8 | 9 | 10 |
|---|---|---|---|---|---|---|---|---|---|
| Deepproblog | 53.53 | 40.42 | 33.67 | 29.86 | 27.51 | 25.46 | 23.63 | 22.52 | 21.47 |
| DeepStochlog | 56.21 | 44.43 | 39.06 | 36.02 | 34.02 | 31.14 | 27.74 | 24.38 | 21.24 |
| NeurASP | 53.66 | 36.07 | 28.39 | 29.77 | 14.63 | 23.25 | 23.61 | 22.51 | 7.63 |
| Ground ABL | 42.50 | 42.00 | 98.25 | 26.00 | 30.25 | 25.25 | 25.75 | 24.00 | 20.25 |
| A3BL | 46.16 | 99.50 | 38.00 | 26.00 | 30.25 | 24.50 | 25.75 | 24.00 | 20.25 |
| $VL_\perp$ | **100.00** | 41.38 | 99.50 | 99.80 | 99.65 | 99.19 | 70.70 | 98.73 | 98.28 |
| $VL_\perp^{TTC}$ | **100.00** | 49.83 | **100.00** | **100.00** | **100.00** | **100.00** | 69.20 | 99.95 | **100.00** |
| $VL_{\not\perp}$ | **100.00** | 99.88 | 99.75 | **100.00** | 99.75 | 99.00 | 99.25 | 97.75 | 48.80 |
| $VL_{\not\perp}^{TTC}$ | **100.00** | **100.00** | **100.00** | **100.00** | **100.00** | 99.95 | **100.00** | **100.00** | 51.40 |

*Table 2.* The comparison of symbol recognition accuracy on the Sort dataset.

| Method | 4 | 5 | 6 | 7 | 8 |
|---|---|---|---|---|---|
| Deepproblog | 96.65 | 97.12 | 95.93 | 39.36 | 45.97 |
| DeepStochlog | 84.27 | 81.72 | 77.28 | MLE | MLE |
| NeurASP | 5.03 | 8.95 | TLE | TLE | TLE |
| Ground ABL | 10.20 | 20.22 | 24.19 | 29.73 | 39.35 |
| A3BL | 29.75 | 69.43 | 49.28 | 49.61 | 97.28 |
| $VL_\perp$ | 77.50 | 77.20 | 96.67 | 68.86 | 98.23 |
| $VL_\perp^{TTC}$ | 78.75 | 76.40 | **99.67** | 69.00 | 99.89 |
| $VL_{\not\perp}$ | 97.00 | 98.76 | 97.97 | 98.20 | 98.71 |
| $VL_{\not\perp}^{TTC}$ | **99.25** | **99.66** | **99.67** | **99.78** | **99.90** |

*Table 3.* The comparison of symbol recognition accuracy on the Match dataset.

| Method | 6 | 7 | 8 | 9 | 10 |
|---|---|---|---|---|---|
| Deepproblog | 16.72 | 14.30 | 12.33 | 11.03 | 10.25 |
| DeepStochlog | 16.68 | 14.28 | 12.70 | 26.17 | MLE |
| NeurASP | 18.82 | 14.02 | 12.20 | 8.68 | TLE |
| Ground ABL | 16.68 | 14.28 | 12.35 | 11.03 | 10.02 |
| A3BL | 16.68 | 14.28 | 12.35 | 11.03 | 10.02 |
| $VL_\perp$ | 65.95 | 42.50 | 73.97 | 97.38 | 96.67 |
| $VL_\perp^{TTC}$ | **68.20** | 44.63 | 72.33 | **99.85** | 99.67 |
| $VL_{\not\perp}$ | 66.27 | 73.25 | 98.22 | 70.35 | 97.08 |
| $VL_{\not\perp}^{TTC}$ | 68.08 | **75.72** | **99.97** | 81.00 | **99.68** |

*Table 4.* The comparison of symbol recognition accuracy on the Chess dataset.

| Method | 2 | 3 | 4 | 5 | 6 |
|---|---|---|---|---|---|
| Deepproblog | 49.46 | 33.10 | 25.08 | 49.46 | 15.66 |
| NeurASP | 53.66 | 36.07 | 23.93 | 19.07 | 18.82 |
| Ground ABL | 49.90 | 67.65 | 75.70 | 70.55 | 33.95 |
| A3BL | **100.00** | 99.80 | 75.95 | 38.30 | 34.35 |
| $VL_\perp$ | **100.00** | **99.90** | 98.00 | 95.70 | 95.15 |
| $VL_\perp^{TTC}$ | **100.00** | **99.90** | 92.55 | 91.30 | 92.55 |
| $VL_{\not\perp}$ | **100.00** | **99.90** | 98.00 | 95.70 | 95.15 |
| $VL_{\not\perp}^{TTC}$ | **100.00** | **99.90** | 92.55 | 91.30 | 92.55 |

network architecture (denoted as $f$) for symbol recognition from $X$ to $S$, with a learning rate of 0.001 and Adam optimizer for optimization. Due to the fact that many algorithms train extremely slowly, in order to conduct a performance comparison with them as comprehensively as possible, we set a unified number of epochs to 10. All experiments were completed on 4 A800 GPUs. The comparison methods include Deepproblog, Deepstochlog, NeurASP, Ground ABL, and A3BL. For algorithms that are only suitable for supervised learning, we used [True, False], i.e., whether the result matches the knowledge base, as the label for the new tasks to enable comparison. Additionally, we compared the performance of four versions of the algorithm in ablation experiments, including two Score settings under the independent and non-independent hypotheses represented by $VL_\perp$ and $VL_{\not\perp}$, to explore the impact of optimization target settings on VL. Under the independent hypothesis, the Score setting used the neural network model's confidence, while under the non-independent hypothesis, the Score used a combination of consistency and confidence. Furthermore, we compared the impact of using test time correction in both cases and used the tag TTC to indicate them. Our knowledge base is in the form of a verification function, and the program was written in simple Python code. During the experiments, NeurASP experienced severe timeouts (no results returned after over 300 hours), and Deepstochlog encountered memory issues (the table size limit in swi-prolog was set to $10^{12}$). In the experimental results, these issues were indicated by TLE (Time Limit Exceeded) and MLE (Memory Limit Exceeded), respectively.

## 7.1. Addition

The experiment on the Addition task is an extension of the addition experiment from Deepproblog, where the answer part was modified to be the input number images. To verify its generalizability, we tested all addition bases between binary and decimal numbers. The results demonstrated the effectiveness of VL on this task, and showed that with only the addition rule, the model could learn to classify all

*Table 5.* The comparison of time consumption on the Addition dataset.

| Method | 2 | 3 | 4 | 5 | 6 | 7 | 8 | 9 | 10 |
|---|---|---|---|---|---|---|---|---|---|
| Deepproblog | 2778.45 | 9118.23 | 15281.67 | 21273.89 | 23988.50 | 30621.92 | 33400.69 | 38804.44 | 42391.21 |
| DeepStochlog | 1289.68 | 454.59 | 984.16 | 1251.83 | 1636.27 | 1583.25 | 2168.25 | 2743.55 | 3097.91 |
| NeurASP | 1133.23 | 1475.02 | 2407.18 | 2354.81 | 4169.20 | 7648.07 | 13483.66 | 23013.77 | 38526.15 |
| Ground ABL | 135.05 | 140.47 | 141.60 | 133.90 | 86.68 | 131.38 | 160.69 | 126.84 | 123.05 |
| A3BL | 187.25 | 183.92 | 200.41 | 203.90 | 202.73 | 161.57 | 184.81 | 185.87 | 190.74 |
| $VL_\perp$ | 112.04 | 114.80 | 111.79 | 111.94 | 111.36 | **109.58** | **106.66** | 112.81 | **116.26** |
| $VL_{\not\perp}$ | **110.48** | **101.17** | **80.32** | **91.05** | **80.79** | 111.18 | 112.27 | **82.16** | 118.36 |

numbers. The results are showed in Table 1.

### 7.2. Sort

The Sort task is an extension of the sort task from Deepproblog, where digit recognition is performed based on an ordered sequence of images. The only knowledge in the knowledge base is the ordering of the numbers. The experiments demonstrated that the model, relying solely on the ordering, could successfully perform recognition for all digits. In the experiments, we tested ordered sequences with lengths ranging from 4 to 8, and the results confirmed the effectiveness of VL under the sorting rule. The results are showed in Table 2.

### 7.3. Match

The Match task is an extension of the anbncn task from Deepstochlog. The task involves character recognition based on fixed model strings in an unsupervised setting. The original task only matched the character set $a^n b^n c^n$, but we increased the task difficulty by extending it to situations not limited to three character sets, with $n$ being variable. We conducted experiments with character categories ranging from 6 to 10 and validated the effectiveness of VL in the character matching task. The results are showed in Table 3.

### 7.4. Chess

The Chess task comes from (Dai et al., 2019), which is an extension of the 8-Queens problem. In the Chess task, there are six types of pieces: the bishop moves diagonally any number of squares, the king moves one square in any direction, the knight moves in an L-shape, the pawn moves one square diagonally forward, the queen moves any number of squares along a straight line or diagonally, and the rook moves any number of squares along a straight line. We identify the type of piece based on the changes in the chessboard configuration. Experiments were conducted with 2 to 6 types of chess pieces, with the piece types dynamically added based on the lexicographical order of the piece names. The results are showed in Table 4.

### 7.5. Time Consumption

In addition to its excellent performance, VL also demonstrates exceptional time efficiency. We conducted a runtime comparison on the addition dataset, with time measured in seconds. The results show that VL even outperforms Ground ABL, which has preprocessing of the knowledge base. This is because VL finds the optimal solution with far fewer verification steps. Compared with the training time of some methods that increases rapidly as the symbol space expands, the growth rate of VL's training time is relatively slow. The results are showed in Table 5.

## 8. Conclusion

This research explores the feasibility of unsupervised Nesy system from both theoretical and practical perspectives. We demonstrate the necessity of developing unsupervised Nesy systems through practical cases, while highlighting three major challenges: reduced symbolic information availability, expanded solution space, and more prevalent shortcut issues.

We propose a verification learning framework that transforms traditional symbolic reasoning starting from a supervision signal Y into a verification paradigm independent of Y. This framework is formalized as a Constraint Optimization Problem, for which we prove that it can be solved with sub-exponential complexity under the condition of independence and the more general condition of monotonicity. We propose a corresponding dynamic combinatorial sorting algorithm for this problem. We also address the exponential growth of shortcuts through distribution alignment.

The theoretical foundation establishes the problem types addressable by VL using group theory, accompanied by generalization error analysis. Experimental validation across four unsupervised learning tasks demonstrates breakthrough progress from infeasible to feasible solutions.

Future work will focus on developing more sophisticated VL frameworks for complex tasks and exploring applications in broader unlabeled data scenarios.

## Acknowledgements

This research was supported by the Key Program of Jiangsu Science Foundation (BK20243012), Leading-edge Technology Program of Jiangsu Science Foundation (BK20232003) and the Fundamental Research Funds for the Central Universities (022114380023).

## Impact Statement

This paper presents work whose goal is to advance the field of Machine Learning. There are many potential societal consequences of our work, none which we feel must be specifically highlighted here.

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

# A. Proof of Theorems

## A.1. Proof of Theorem 4.1

We can use proof by contradiction. Suppose there exists a score-ranked $S_i$ at position $i$ such that $S_i \notin \{S_1, \ldots, S_{i-1}\}$, and there does not exist $0 < j < i$, $0 < p \le l$, $0 < q < k$ such that modifying the symbol $S_{p,q}$ at position $p$ of $S_j$ to $S_{p,q+1}$ gives $Suc(S_j)_p$ with $Score(Suc(S_j)_p) \ge Score(S_i)$. Then for any $0 < j < i$, there are at least two positions $p_a$ and $p_b$ where the symbols $S_{p_a,q_{ai}}$, $S_{p_b,q_{bi}}$ of $S_i$ at positions $p_a$ and $p_b$ and the symbols $S_{p_a,q_{aj}}$, $S_{p_b,q_{bj}}$ of $S_j$ at these positions satisfy $q_{ai} \ge q_{aj} + 1$, $q_{bi} \ge q_{bj} + 1$; or there is at least one position $p_a$ where the symbol $S_{p_a,q_{ai}}$ of $S_i$ and $S_{p_a,q_{aj}}$ of $S_j$ satisfy $q_{ai} \ge q_{aj} + 2$.

For the first case, there must exist $1 < j < i$ such that modifying the symbol at position $p_a$ (similarly for $p_b$) of $S_j$ to $S_{p_a,q_{aj}+1}$ gives $Suc(S_j)_{p_a}$. Since $q_{ai} \ge q_{aj} + 1$, $q_{bi} > q_{bj}$, and for any other position $c \notin \{a, b\}$, $q_{ci} \ge q_{cj}$, and since score is a sorted probability, we have:

$$score(S_{p_a,q_{ai}}) \ge score(p_a, q_{aj} + 1), \quad score(S_{p_b,q_{bi}}) \ge score(p_b, q_{bj}), \quad score(S_{p_c,q_{ci}}) \ge score(p_c, q_{cj})$$

By the independence assumption, $Score(S) = \prod_{i=1}^{l} score(p_i, q_i)$, so:

$$Score(Suc(S_j)_{p_a}) \ge Score(S_i)$$

which leads to a contradiction, thus the assumption is invalid.

For the second case, there must exist $1 < j < i$ such that modifying the symbol at position $p_a$ of $S_j$ to $S_{p_a,q_{aj}+1}$ gives $Suc(S_j)_{p_a}$. Since $q_{ai} > q_{aj} + 1$ and for any other position $c \ne a$, $q_{ci} \ge q_{cj}$, we have:

$$score(S_{p_a,q_{ai}}) \ge score(p_a, q_{aj} + 1), \quad score(S_{p_c,q_{ci}}) \ge score(p_c, q_{cj})$$

Thus:

$$Score(Suc(S_j)_{p_a}) \ge Score(S_i)$$

leading to a contradiction, so the assumption is invalid. Therefore, Theorem 4.1 is proved.

## A.2. Proof of Theorem 4.2

The proof process is the same as that of Theorem 4.1. Due to the satisfaction of the monotonicity assumption, for any $0 < p \le l$ and any $0 < q < k$, there does not exist $S'$ such that modifying $s_p$ from $s_{p,q}$ to $s_{p,q+1}$ yields $Suc(S')_p = (s_1, \ldots, s_{p-1}, s_{p'}, s_{p+1}, \ldots, s_l) \in Suc(S')$ satisfying $Score(Suc(S')) > Score(S')$. It can be seen that for any $q' > q$, modifying $s_p$ from $s_{p,q}$ to $s_{p,q'}$ to obtain $S'_q$ also cannot satisfy $Score(Suc(S'_q)) > Score(S')$.

Thus, for the first case, where $q_{ai} \ge q_{aj} + 1$, $q_{bi} > q_{bj}$, and for any other position $c \notin \{a, b\}$, $q_{ci} \ge q_{cj}$, it can similarly be proven that $Score(Suc(S_j)_{p_a}) \ge Score(S_i)$ holds, leading to a contradiction and invalidating the assumption.

For the second case, where $q_{ai} > q_{aj} + 1$ and for any other position $c \ne a$, $q_{ci} \ge q_{cj}$, it can similarly be proven that $Score(Suc(S_j)_{p_a}) \ge Score(S_i)$ holds, leading to a contradiction and invalidating the assumption.

Therefore, Theorem 4.2 is proved.

## A.3. Proof of Theorem 4.4

The sufficiency can be proved according to Theorem 4.2.

When the monotonicity condition is not satisfied, even if we find the top-ranked $S_1$ by Score, we still need to calculate the scores of all $S' \in \mathcal{S}$. It is impossible to find the maximum solution without exhaustive traversal, and even more impossible to infer the $i_{th}$ largest solution from the first $i - 1$ ranked solutions. Thus, the necessity is proved.

## A.4. Proof of Theorem 6.1

First, consider the clustering ability of the function $f$. We define the clustering error as the error rate between the labels assigned to samples under the optimal label assignment strategy after clustering and the true labels. According to the Rademacher complexity theory, for any function $f \in \mathcal{F}$ and a $\rho$-Lipschitz continuous loss function $L$, under the action of

$n$ samples, the generalization error rate $R_{\text{oracle}}(f)$ under the optimal label assignment strategy can be determined by the empirical error rate $\hat{R}(f)$ and the model space complexity $\mathcal{R}_n(\mathcal{F})$, that is, with probability at least $1 - \delta$, it holds that:

$$R_{\text{oracle}}(f) \le \hat{R}(f) + 2\rho\mathcal{R}_n(\mathcal{F}) + 3\sqrt{\frac{\log(2/\delta)}{2n}}$$

Due to the indistinguishability of symbols in the system, the error between the true label assignment strategy and the optimal label assignment strategy differs by at most $R_{\text{task}}^{\text{up}}$. The system's generalization error $R(f)$ is determined by the generalization error $R_{\text{oracle}}(f)$ under the optimal label assignment strategy and the maximum difference $R_{\text{task}}^{\text{up}}$ between the actual label assignment strategy and the optimal one, i.e.,

$$R(f) \le R_{\text{oracle}}(f) + R_{\text{task}}^{\text{up}}$$

Finally, it can be concluded that with probability at least $1 - \delta$:

$$R(f) \le \hat{R}(f) + 2\rho\mathcal{R}_n(\mathcal{F}) + 3\sqrt{\frac{\log(2/\delta)}{2n}} + R_{\text{task}}^{\text{up}}$$

## B. The Programs of All of the Verification Function

```
# Addition
def digits_to_number(digits,num_classes=2):
    number = 0
    for d in digits:
        number *= num_classes
        number += d
    return number

def number_to_digits(number, digit_size,num_classes=2):
    digits=[]
    for i in range(digit_size):
        digits.append(number%num_classes)
        number//=num_classes
    return digits[::-1]

def V_KB(nums,num_digits,num_classes):
    nums1,nums2,nums3=nums[:num_digits],nums[num_digits:num_digits*2],\
    nums[num_digits*2:]
    return (digits_to_number(nums1,num_classes=num_classes) \
    + digits_to_number(nums2,num_classes=num_classes)==\
    digits_to_number(nums3,num_classes=num_classes))

# Sort
def V_KB(nums):
    l=len(nums)
    for _ in range(l-1):
        if nums[_+1]<=nums[_]:
            return False
    return True

# Match
def V_KB(nums):
    l=len(nums)
```

```
    count=None
    cur_count=0
    for _ in range(l):
        if _ >0 and nums[_]<nums[_-1]:
            return False
        elif _>0 and nums[_]>nums[_-1]:
            if count is None:
                count=cur_count
            elif count!=cur_count:
                return False
            cur_count=0
        cur_count+=1
    return count is None or cur_count==count

# Chess
def attack(type, x1, y1, x2, y2):
    if type == 0:
        return bishop_attack(x1, y1, x2, y2)
    elif type == 1:
        return king_attack(x1, y1, x2, y2)
    elif type == 2:
        return knight_attack(x1, y1, x2, y2)
    elif type == 3:
        return pawn_attack(x1, y1, x2, y2)
    elif type == 4:
        return queen_attack(x1, y1, x2, y2)
    elif type == 5:
        return rook_attack(x1, y1, x2, y2)
    return False

def king_attack(x1, y1, x2, y2):
    # King moves one step in any direction
    return abs(x1 - x2) <= 1 and abs(y1 - y2) <= 1

def queen_attack(x1, y1, x2, y2):
    # Queen moves straight or diagonal
    return self.straight_attack(x1, y1, x2, y2) or \
    self.diagonal_attack(x1, y1, x2, y2)

def rook_attack(x1, y1, x2, y2):
    # Rook moves straight
    return self.straight_attack(x1, y1, x2, y2)

def bishop_attack(x1, y1, x2, y2):
    # Bishop moves diagonally
    return self.diagonal_attack(x1, y1, x2, y2)

def knight_attack(x1, y1, x2, y2):
    # Knight moves in an "L" shape
    return (abs(x1 - x2) == 2 and abs(y1 - y2) == 1) or \
    (abs(x1 - x2) == 1 and abs(y1 - y2) == 2)

def pawn_attack(x1, y1, x2, y2):
    # Pawn attacks diagonally (assuming it's a white pawn)
```

```
    return abs(x1 - x2) == 1 and y2 - y1 == 1

def straight_attack(x1, y1, x2, y2):
    # Moves straight: either same row or same column
    return x1 == x2 or y1 == y2

def diagonal_attack(x1, y1, x2, y2):
    # Diagonal move: difference between x and y is the same
    return abs(x1 - x2) == abs(y1 - y2)

def V_KB(type, pos):
    l=len(type)
    for i in range(l):
        for j in range(i+1,l):
            if attack(type[i],pos[i][0],pos[i][1],pos[j][0],pos[j][1]):
                return True
    return False
```

All of the code is open-sourced on the github https://github.com/VerificationLearning/VerificationLearning.

