# OpenReview forum: "Verification Learning: Make Unsupervised Neuro-Symbolic System Feasible"
_ICML.cc/2025/Conference — ICML 2025 poster_

### Official Review · Reviewer_W1W6 · 2025-03-01

**Overall Recommendation:** 2

**Summary:**

This paper introduces a novel learning paradigm, named verification learning, which transform the label-based reasoning process in neuro-symbolic into a label-free verification process. It achieves good learning results solely by relying on unlabeled data and a function that verifies whether the current prediction conform to the rules. The theoretical analysis points out which tasks in neuro-symbolic systems can be completed without labels and explains why rules can replace infinite labels.

**Claims And Evidence:**

Yes.

**Essential References Not Discussed:**

No.

**Experimental Designs Or Analyses:**

Yes.

**Methods And Evaluation Criteria:**

Yes.

**Other Comments Or Suggestions:**

See questions below.

**Other Strengths And Weaknesses:**

**Strength**: \
(1) The idea and motivation is clear state. The writing is good and the method pipeline is easy to follow. \
(2) This paper provides a comprehensive theoretical foundation for the proposed method.

**Weaknesses**: \
(1) The novelty of the proposed method requires further elaboration and discussion. \
(2) The experiments is somewhat weak and necessitates further expansion and refinement to ensure robustness and comprehensiveness.

**Questions For Authors:**

(1) The proposed method seems simple, and I concern about the novelty. I recognize the significance of a label-free neuro-symbolic framework and the urgency of developing corresponding solutions. While I appreciate the idea and motivation behind this paper, I remain interested in how other reviewers evaluate its novelty. Could the authors provide a more detailed explanation of the proposed method's novelty from both algorithmic and theoretical perspectives? \
(2) The algorithm description requires further refinement, as I believe many details are omitted and not adequately explained in the supplementary materials. For example, what is the precise definition of the score in Equation 1? Additionally, are there brief proofs provided for the theorems and propositions presented in the paper? \
 (3) The experiments are weak incomplete; additional evaluation metrics and baselines are necessary to facilitate more comprehensive comparisons.

**Relation To Broader Scientific Literature:**

Neuro-Symbolic Learning, Constraint optimization Problem.

**Theoretical Claims:**

Yes.

---

> ### Author Rebuttal · Authors · 2025-04-01
>
> Dear Reviewer W1W6:
>
> Thank you for valuable comments.
> ### Regarding Question 1 and Weaknesses 1:
> Methodological Innovations:
> 1. Explored fully unsupervised Nesy algorithms and provided a systematic solution.
> 2. Proposed a universal approach for finding globally optimal solutions to Nesy search problems solvable in sub-exponential time.
> 3. Demonstrated that in the Nesy domain, a single validation function can replace an entire complex knowledge base.
> 4. Shifted the paradigm from inference to validation, expanding the symbolic space complexity that can be solved.
>
> Theoretical Innovations:
> 1. Proved that when a symbolic system is sufficiently differentiated, rules can theoretically replace all labels. This is equivalent to proving that rule-based clustering alone can achieve classification.
> 2. Demonstrated that monotonicity is a necessary and sufficient condition Nesy to achieve a globally optimal solution in sub-exponential time.
> 3. Established that the irreplaceability of symbols within a system is a necessary condition for their correct recognition.
>
> ### Regarding Question 2:
> Regarding Formula (1):
> In Formula (1), we have generalized various algorithms, such as deepproblog and others, to highlight the commonality in their form. Therefore, the Score does not refer to any specific score. I outline the differences in the Score chosen by different algorithms for comparison.
> - In deepproblog and deepstochlog, $Score(S)=\prod_{i=1}^{|path|} p(path_i)$ where path refers to the path leading to S derived from the root.
> - In ABL, $Score(S)=\sum_{i=1}^{m}[f(x_i)==s_i]$ known as the Consistency, which represents the number of symbols in S that align with the predictions.
> - In WSABL, $Score(S)=\prod_{i=1}^{m} confidence(x_i)_{s_i}$ known as the Confidence, which represents the probability that predicts.
> - In VL. In the independent mode, we use the Confidence. In the non-independent mode, Score(S)=(Consistency(S), Confidence(S)). This means seeking the solution with the highest consistency and, the one with the highest confidence among them.
>
> Regarding Proofs
>
> For Theorem 4.1, we use proof by contradiction. Suppose for the i-th largest solution i>1, there is no 0<j<i such that i=suc(j). Then, there must exist some l where i=suc(l) and score(l)>score(i). Since no 0<j<i satisfies i=suc(j), it follows that l>i, contradicting score(l) >score(i). Thus for any i>1, there must exist 0<j< i such that i=suc(j).
>
> The proof for Theorem 4.2 follows the same reasoning as Theorem 4.1.
>
> For Proposition 4.3, if S is independent in terms of Score, then Score(S) is the product of individual scores. This ensures that reordering any subset does not affect the relative ranking of scores, making independence a sufficient condition for monotonicity.
>
> For Theorem 6.1, let $\hat{R}(f)$ be the empirical loss of the best permutation for clustering. The difference between its empirical and generalization error is bounded by
> $2\rho \mathcal{R}_n(F)+3 \sqrt{\frac{\log(2/\delta)}{2n}}$.
>
> The difference between the optimal and true permutation is given by $R^{up}_{task}$.
> ### Regarding Weaknesses 2 and Question 3:
> We expanded the experiments by adding 4 new comparison methods. These include 2 classic methods, Hybrid and ABL-zoopt, where Hybrid is an algorithm for solving the CSP problems using neural networks, and ABL-zoopt is a classic method based on gradient-free optimization. Additionally, we included 2 recent methods, REFL and PSP. REFL accelerates the inference process using the Reflection mechanism which is the best paper at AAAI 25. PSP is the latest algorithm that combines probabilistic symbolic perception with symbolic context.
> |Method|2|3|4|5|6|7|8|9|10|
> |-|-|-|-|-|-|-|-|-|-|
> |Hybrid|53.75|40.85|38.73|35.48|28.38|30.48|24.13|21.83|21.43|
> |ABL-Zoopt|100.00|44.50|99.55|70.33|40.15 |25.25|26.43|48.25|48.03|
> |REFL|99.98|42.60|64.80|99.43|99.05|62.78|57.57|62.93|49.08|
> |PSP|99.88|45.58|64.75|99.03 |75.38|67.58|63.30|98.23 |97.38|
> |VL|100.00|49.83|100.00|100.00|100.00|100.00|69.20|99.95|100.00|
>
> Additionally, we have included more evaluation metrics. In addition to the accuracy in symbol recognition, we have also included classic metrics such as Recall, Precision, F1 Score, and ROC-AUC. Furthermore, we have added the Global Accuracy metric, which describes the probability that the network provides the correct answer for all symbols in a given set. It is used to reflect the algorithm's ability to find the precise solution.
> |Method|Accuracy|Recall|Precision|F1|ROC-AUC|Global Accuracy|
> |-|-|-|-|-|-|-|
> |Deepporblog |21.41|2.14|10.00|3.53|50.00|0.75|
> |Deepstochlog|12.50|1.25|10.00|2.22|47.60|0.00|
> |NeurASP|7.44|1.23|9.96|1.55|49.41|0.00|
> |ABL|21.43|2.14|10.00|3.53|50.00|1.50|
> |WSABL|21.47| 21.47|10.00|3.54|50.73|0.98|
> |Hybrid|21.43|2.14|10.00|3.53|50.00|1.50|
> |ABL-Zoopt|48.03|22.27|27.23|24.75|81.23|5.20|
> |REFL|49.09| 27.29|30.93|29.91|78.30|10.90|
> |PSP|97.38| 97.09|96.54| 96.77|99.92|89.80|
> |VL|99.95|99.93|99.94|99.93|99.98|99.90|

---

### Official Review · Reviewer_Vjyz · 2025-03-12

**Overall Recommendation:** 3

**Summary:**

In this work, the authors introduce a novel learning paradigm called Verification Learning (VL) to address the problem of learning without labels in neuro-symbolic models. VL corresponds to flipping the standard label-based reasoning [S,KB |= Y] into predicting a set of possible candidate solutions leveraging the knowledge base and the labels [KB,Y |= candidates(S)], and eventually remove the labels from the picture, to infer the set of candidates directly from the knowledge base [KB |= candidates(S)]. Since directly inferring the correct candidate assignment from the set of candidate symbols is computationally prohibitive (would require enumerating all the possible candidates, infeasible in practical scenarios) the author resorts to a generation-validation strategy. To be competitive, this verification approach exploits the assumption that the generated candidates can be sorted according to some scores (that matches the fitness of the candidates), allowing the exploration of the candidate space in an efficient manner using a heap structure. The authors show that this is possible not only when the score factorizes over the symbols (independent assumption, Th. 4.1) but also in the more relaxed constraint where the score satisfies monotonicity (Th. 4.2). The authors also introduce a distribution alignment with some prior to avoid shortcut solutions. Finally, the authors include a theoretical analysis on the upper bounds on accuracy that VL can achieve and experimental results on 4 tasks: addition, sort, match, chess.

## update after rebuttal
I acknowledge and appreciate the efforts made by the authors, and I have carefully reviewed their responses. The additional experimental results on the Road-R dataset, the ablation on the distribution alignment, the additional baselines introduced for other reviewers (Hybrid, ABL-zoopt, REFL, PSP), and the empirical quantification of the benefits of sorting algorithms increased my confidence in the value of this work. I also want to thank the authors for addressing my questions and some of the weaknesses highlighted in the initial review. Even if I agree with Reviewer W1W6 on the fact that the work still requires further improvement, I am raising my score in light of the new experimental results and explanations added during the rebuttal.

**Claims And Evidence:**

The authors claim that VL can be a valid alternative to the standard label-based reasoning to learn symbols in neuro-symbolic models. This is supported by presenting (loose) theoretical upper bounds motivated by group theory and a limited set of empirical results on four different symbolic tasks (addition, sort, chess, and match). A second claim is that VL can reduce known issues of neuro-symbolic models, i.e., label leakage and shortcut learning, which comes by the definition of the methodology itself (labels are not used in the training process at all). The authors claim that the proposed sorting strategy gives an edge compared to a naive implementation that evaluates of all the possible symbol assignments. Clearly this is the case, but the advantage is not quantified experimentally. Also, while the argument about monotonicity as necessary condition to enable an efficient combinatorial sorting solution, it is not clear how often and for which score functions this condition would generally hold. Finally, the extent to which this methodology could be apply in real-world scenarios (for which collecting labels can be expensive) is not clear, as a complete verifier would be always needed to allow the generation-verification procedure. Adding an outlook on how this methodology could scale to more complex scenarios could help.

**Essential References Not Discussed:**

I am not aware of essential references related but not discussed in the manuscript.

**Experimental Designs Or Analyses:**

The methodology of the experiments is not sufficiently clear and could benefit from some more careful rewriting. For instance, it is not clear what the accuracies reported in the table represent, how exactly is the model trained, how the baseline models compare to the proposed VL framework in terms of size/supervision/etc. Moreover, some of the novelties introduced in this work are not really ablated in the experimental analysis. For instance, there is no quantification of the computational benefit of the efficient combinatorial sorting solution and no ablation of the distribution alignment technique introduced in Section 5 (which also comes out a bit of nowhere).

**Methods And Evaluation Criteria:**

The proposed benchmark are valid choices to evaluate the claims, even if slightly simplistic. This does not represent a major issue, since it is in line with the datasets used to benchmark standard NeSy models. The proposed methodology is principled and, to the best of my knowledge, is original compared to previous works.

**Other Comments Or Suggestions:**

-	In Line 45, page 1 you use a comma to “tie” together the two symbols, while in Line 51 you use a plus symbol (not clear if this is intentional or not).
-	KL in Line 70 of page 2 left column pops out of nowhere.
-	The references should be revised, as some are missing quite important fields (e.g. Yang et al. He et al., and van Krieken et al are all missing the year).
-	The last sentence of the paragraph, line 126 left column, does not sound correct.
-	Line 133, page 3 right column there is a comma instead of a full stop.
-	Lines 174—195 inconsistent use of ; and .

**Other Strengths And Weaknesses:**

A major strength of this paper is the originality of the idea, which is interesting and allowed by a series of non-trivial steps, e.g. the efficient combinatorial sorting to speed up the COP. However, the presentation of the material could be greatly improved. I believe the manuscript still needs to be ironed in many parts, the notation needs to be simplified in some sections, and less background knowledge must be given for granted in others. Some examples:
-	The examples in the 3rd paragraph of the introduction are not well introduced and it is borderline impossible to understand them without referring to the original papers. More in general, I think that giving for granted a lot of background knowledge is an issue underlying many parts of the text, which undermines its accessibility and utility for the community in general.
-	Sometimes the notation is heavy, overlapping, and confounding. For instance, take the paragraph in Line 206 left column: on its own the concept it’s rather straightforward, but the imprecise notation (S_{i+1}\in S_j, while it should be S_{i+1}\in Suc(S_j)) considerably complicates the understanding of it.
-	It is not explained in the manuscript what are the test time corrections (TTC) mentioned in the results section.
-	Tables are very poorly commented and could benefit from more explicative captions or more detailed descriptions in the main text. Right now, they are not self-contained, and one should go through the entire result section to understand what the different columns represent. The tables do not report which metric is used in the corresponding experiments.

**Questions For Authors:**

-	In the 1st example of section 6, you state that the error upper bound for Sudoku is 100%, because any permutation of numbers within a row, column or grid satisfies the constraints within them. However, these permutations would not satisfy the constraints of other entities (e.g., permuting within a row would still guarantee the satisfaction within itself, but would most likely result in breaking the constraints of other columns/grids). Why are these not taken in account?
-	Where is the L (learning) in VL? What is effectively trained in your setup? This should be really clarified in the manuscript, which right now is missing a clear outline of the proposed system and training pipeline.

**Relation To Broader Scientific Literature:**

This work is mainly related to works on label leakage (Chang 2020) and shortcut learning (Yang 2024, He 2024) as a potential solution to these problems (labels are completely removed from the learning process, hence no leakage or shortcut is possible)

**Theoretical Claims:**

The authors propose three main theoretical contributions in the paper:
-	A proof that, when the symbols satisfy the independence assumption, the exploration of the candidates can be performed efficiently “exploding” a limited number of successors for each candidate and guiding the search using a heap structure [Th 4.1].
-	A proof that independence is a sufficient but not necessary condition. On the other hand, the relax constraint of monotonicity of the scoring function is a more relaxed but necessary condition [Th 4.2].
-	Theoretical bounds on which problem types are addressable by the VL framework using group theory [Section 6].
The first two are correct, I did not check carefully the third.

---

> ### Author Rebuttal · Authors · 2025-04-01
>
> Dear Reviewer Vjyz:
>
> Thank you for valuable comments.
> ### Regarding the Benefits of Sorting Algorithms
> We supplement this comparison with the time consumption of the naïve sorting method. As the space grows exponentially, the time required by the naïve method increases tenfold, while DCS remains stable.
> |Algorithm|2|3|4|5|6|7|8|9|10|
> |-|-|-|-|-|-|-|-|-|-|
> |VL|112|114|111|111 |111|109|106|112|116|
> |Naive|114|127|173|213|239|394|573|641|1254|
>
> ### Regarding Monotonicity
> When monotonicity is not satisfied, it can be proven that no algorithm with a complexity lower than exponential can find the optimal solution. Fortunately, the optimization objectives in current Nesy frameworks almost satisfy monotonicity.
>
> 1.Metrics that satisfy independence
> An example is the Confidence Score, defined as:
> $Confidence(S)=\prod_{i=1}^{m} confidence(x_i)_{s_i}$
>
> 2.Metrics that do not satisfy independence but still satisfy monotonicity.
> $Consistency(S)=\sum_{i=1}^{m}\left[ f(x_i)=s_i\right]$
>
> 3.Combinations of Metrics
> Any metric formed by adding, multiplying (with positive values), or concatenating also retains monotonicity. For example, the two-dimensional tuple: $(Consistency(S), Confidence(S))$ remains a monotonicity-preserving metric.
> ### Regarding Practical Applications
> In applications, VL can be applied in various scenarios. In autonomous driving, a vehicle can automatically recognize pedestrians, traffic lights, and crosswalks in an image using only the traffic rules. In circuit fault analysis, VL can identify faulty components based solely on input circuit diagrams and electrical rules.
> We further validated VL in real-world scenarios by conducting experiments on the Road-R autonomous driving dataset with driving rules.
> |Method|4|5|6|7|8|9|
> |-|-|-|-|-|-|-|
> |Deepporblog|6.24|16.91|20.31|7.46|12.57|11.50
> |Deepstochlog|5.98|19.12|MLE|MLE|MLE|MLE|
> |NeurASP|27.30|22.34|6.12|TLE|TLE|MLE|
> |ABL|86.58|58.01|44.93|28.15|19.50|5.18|
> |WSABL|87.22|57.38|32.11|31.01|36.29|22.45|
> |$VL_{\not\perp}$|92.06|86.54|86.53|82.16|82.90|73.36|
> |$VL_{\not\perp}^{TTC}$|93.43|92.38|93.87|90.63|93.33|88.48|
> |$VL_{\perp}$|92.06|87.84 |87.59|84.93|82.80|72.72|
> |$VL_{\perp}^{TTC}$|93.43|92.40|92.93|90.86|91.90|82.24|
> ### Regarding Ablation Study on Alignment
> Without distribution alignment, unsupervised learning can effectively cluster similar samples together but struggles to associate them with the correct symbols.
> |Method|2|3|4|5|6|7|8|9|10|
> |-|-|-|-|-|-|-|-|-|-|
> |$VL_{\not\perp}^{noalign}$|46.25|100.00|34.5|29.6|27.00|25.25|21.83|24.13|21.42|
> |$VL_{\not\perp}$|100.00|99.88|99.75|100.00|99.75|99.00|99.25|97.75|48.80|
> |$VL_{\perp}^{noalign}$|46.60|32.68|29.58|29.60|27.00|35.08|24.13|28.43|31.43|
> |$VL_{\perp}$|100.00|41.38|99.50|99.80|99.65|99.19|70.70|98.73|98.28|
>
> ### Regarding Experimental Designs or Analyses
> The goal of this work is to train neural networks without any labels by utilizing verification functions. This enables the neural networks to accurately extract symbols from raw data. All accuracy values reported in this paper correspond to the accuracy of the neural network in symbol recognition. In Section 7, we introduce the general experimental setup. A LeNet backbone was employed in all cases, with training conducted for 10 epochs.
>
> ### Regarding Question 1
> In this paper, all the permutations we describe are global permutations applied to the entire symbol system. The permutation we aim to express is holistic (in a 9x9 grid) and refers to symbol permutations on the entire Sudoku board. For example, by permuting all the 1s in the 9x9 grid to 2s, 2s to 3s, ..., and 9s to 1s, this global permutation will cause corresponding changes in each row, column, and 3x3 grid, but will not violate any of the constraints on any of them. For example, in a vision Sudoku task with 81 unlabeled images, a completely unsupervised approach would fail to prevent the images from being interpreted as permuted labels. Due to the possibility of all labels being permuted, the error bound caused by the Sudoku task itself can reach 100%.
>
> ### Regarding Question 2
> In Nesy, machine learning models such as neural networks play a role in recognizing abstract symbols from raw data. Our work is no different in this regard, and our ultimate goal is to complete the neural network training process using only a verification function as the knowledge base.
> The pipeline described in this paper can be summarized as follows:
> 1. A set of samples $X=[x_1,\dots,x_m]$ is input into the network f.
> 2. The neural network outputs probability predictions for this set of samples g(X) and label predictions f(X).
> 3. The probability predictions g(X) are then adjusted through distribution alignment to obtain $g_{align}(X)$.
> 4. A Score is chosen as an optimization goal with monotonicity.
> 5. The DCS is called to obtain the optimal result $Y^{val}$ validated by the verification function.
> 6. The result $Y^{val}$ is used as supervision information, and the loss is computed by comparing it with g(X).

---

### Official Review · Reviewer_3PLJ · 2025-03-13

**Overall Recommendation:** 3

**Summary:**

The paper introduces verification learning, a neuro-symbolic paradigm to overcome reliance on labeled data by converting traditional symbolic reasoning into a label-free verification process. VL frames the learning task as a constraint optimization problem and leverages a dynamic Combinatorial sorting algorithm to efficiently find optimal solutions. A distribution alignment method is also introduced to mitigate shortcut issues common in unsupervised settings. Experiments demonstrate the efficacy of VL across various tasks, including addition, sorting, matching, and chess.

**Claims And Evidence:**

Yes

**Essential References Not Discussed:**

No

**Experimental Designs Or Analyses:**

Yes

**Methods And Evaluation Criteria:**

Yes

**Other Comments Or Suggestions:**

1. Clarify explicitly the conditions under which monotonicity fails. Provide a simple example scenario.
2. The readability of certain theoretical sections (especially Sections 4.1 and 4.2) could be improved by using concrete examples to illustrate key concepts clearly.

**Other Strengths And Weaknesses:**

Strengths:
1. The proposed verification learning paradigm is innovative. It clearly addresses critical bottlenecks in existing neuro-symbolic frameworks by eliminating the need for labeled data.
2. The paper provides theoretical analyses that clarify task solvability conditions and generalization bounds.
3. Experiments across diverse tasks demonstrate performance improvements and data efficiency.

Weaknesses:
1. The assumption of monotonicity, although relaxed compared to independence, still limits the generalizability of the proposed DCS algorithm to more complex, non-monotonic scenarios.
2. The theoretical framework, while insightful, may be overly idealized, and the translation of these theoretical insights to complex real-world scenarios may not be straightforward.
3. The complexity of the distribution alignment strategy may not be trivial in tasks with highly imbalanced or unknown natural symbol distributions.
4. In general, this method requires substantial domain knowledge, which may be impractical in many real-world scenarios.

**Questions For Authors:**

1. Can the DCS algorithm handle tasks with high-dimensional symbol spaces or extensive solution spaces without significant computational overhead?
2. In practice, how sensitive is VL to incorrect or noisy verification functions, and how might such imperfections affect learning performance?
3. Could distribution alignment be adapted or automated for scenarios where the natural distribution is not unknown?

**Relation To Broader Scientific Literature:**

This work provides practical and theoretical foundations for unsupervised learning without labelled data.

**Theoretical Claims:**

Yes

---

> ### Author Rebuttal · Authors · 2025-04-01
>
> Dear reviewer 3PLJ:
>
> Thank you for valuable comments.
> ## Weakness 1, Suggestions 1 & 2
> First, we need to clarify that in Nesy, for more complex scenarios where monotonicity is not satisfied, **no algorithm can find the global optimal solution in sub-exponential time. Monotonicity is both a necessary and sufficient condition for solving it**.
>
> In a Nesy problem, where there are **m symbols**, each with **n possible values**, the complexity of the solution space is **$O(n^m)$**. The goal is to find the optimal solution among them.
> - If monotonicity holds, the DCS algorithm can solve the problem efficiently with extremely low complexity.
> - If monotonicity does not hold, the only way to guarantee finding the optimal solution is by exhaustively searching the entire solution space, because even if **$n^m-1$** solutions have already been evaluated, there is no way to determine whether the last remaining solution is optimal.
>
> Example 1: Independence Holds
> For m = 2, n = 2 suppose:
>
> score=[[0.4,0.6],[0.7,0.3]]
>
> Score(s₁=v₁,s₂=v₁)=0.28
> Score(s₁=v₁,s₂=v₂)=0.12
> Score(s₁=v₂,s₂=v₁)=0.42
> Score(s₁=v₂,s₂=v₂)=0.18
>
> Example 2: Monotonicity Holds
>
> Score(s₁=v₁,s₂=v₁)=0.25
> Score(s₁=v₁,s₂=v₂)=0.18
> Score(s₁=v₂,s₂=v₁)=0.34
> Score(s₁=v₂,s₂=v₂)=0.23
> monotonicity is satisfied:
> Score(s₁=v₁,s₂=x)<Score(s₁=v₂,s₂=x)
> Score(s₁=x,s₂=v₂)<Score(s₁=x,s₂=v₁)
>
> Example 3: Monotonicity Does Not Hold
>
> Score(s₁ = v₁,s₂= v₁)=0.23
> Score(s₁ = v₁,s₂ = v₂)=0.25
> Score(s₁ = v₂,s₂ = v₁)=0.34
> Score(s₁ = v₂,s₂ = v₂)=0.18
> Here, fixing one symbol’s value leads to inconsistent ranking of the other symbol’s scores, meaning monotonicity is violated. In this case, there is no alternative to brute-force enumeration to guarantee finding the optimal solution.
> ## Weakness 2
> Our theory is actually applicable.  It demonstrates that in many real-world scenarios—such as **traffic control, industrial anomaly detection, and circuit analysis**—it is entirely feasible to replace all labels with rules for learning.
> Our theory also highlights that the key to replacing labels with rules lies in ensuring that different symbols within the rule system possess irreplaceability. For instance, in autonomous driving tasks, the rule **"Red and green lights cannot be on simultaneously"** cannot distinguish between them. However, the rule **"Stop at red, go at green"** effectively differentiates them. This insight is valuable for **selecting meaningful rules** when constructing a knowledge base.
> ## Weakness 3 & Question 3
> The primary role of distribution alignment is to initialize the neural network’s predictive distribution, preventing it from falling into trivial solutions.
> According to Bayes' theorem, the posterior probability learned by the model can inherently correct biased priors. When the true natural distribution is unknown, a uniform distribution can be used as the prior to ensure a sufficiently diverse initialization. This allows the model to fully utilize all available knowledge and gradually converge to the true distribution.
> Below is a comparison between using a uniform prior for alignment and no alignment on an imbalanced addition dataset:
> |Method|2|3|4|5|6|7|8|9|10|
> |-|-|-|-|-|-|-|-|-|-|
> |No prior|46.60|32.68|29.58|29.60|27.00|35.08|24.13|28.43|31.43|
> |Inexact prior|100.00|100.00|99.95|100.00|58.50|99.90|59.23|49.43|43.45|
>
> ## Weakness 4
> Our algorithm is practical as it only requires writing a verification function, significantly simplifying the complexity of knowledge compared to others.
> Furthermore, when dealing with a large amount of unlabeled data, writing a verification program is more cost-effective than manually labeling. VL can achieve 92.02% based on the autonomous driving rule set Road-R.
>
> ## Question 1
> As the size of the symbol space increases, VL exhibits a low growth in runtime. In Table 2, we compare the runtime as the number of symbol categories increases from 2 to 10, where VL's runtime only increases from 112.04 to 116.26, whereas the runtimes of DeepProbLog increase by a factor of 20.
> This efficiency is due to our sorting algorithm, which ensures that the first solution passing verification is the optimal one. The time complexity is O(K(logK+mlogm+nlogn)), where the constant K represents the position of the first verified solution.
>
> ## Question 2
> We conducted experiments to validate the robustness. In our tests, the verification function returns the correct validation result with probability p, and we gradually decrease p from 100% down to 50%(completely random).
> Our findings indicate that the algorithm's performance only experiences a significant decline when predictions approach complete randomness. As long as the probability of returning correct results is higher than that of returning incorrect results, VL can largely mitigate the impact of errors, demonstrating a strong fault tolerance.
> |p|100|95|90|85|80|75|70|65|60|55|50
> |-|-|-|-|-|-|-|-|-|-|-|-|
> |Acc|98.28|97.58|97.24|94.19|95.02|94.72|93.99|93.55|92.58|83.34|53.58|

---

### Official Review · Reviewer_NuWZ · 2025-03-13

**Overall Recommendation:** 2

**Summary:**

This paper presents a new framework in unsupervised NeSy, excelling in theory and experiments.  And the new framework shows excellent performance across diverse tasks. However, there are several weaknesses that need to be addressed to enhance the quality of the paper. The authors should provide further explanations for the experiment and refine the language clarity.

**Claims And Evidence:**

Yes

**Essential References Not Discussed:**

Yes

**Experimental Designs Or Analyses:**

Yes

**Methods And Evaluation Criteria:**

Yes

**Other Comments Or Suggestions:**

No

**Other Strengths And Weaknesses:**

weakness：
1. The language is occasionally unclear, e.g., "KL paradigm" in the introduction should likely be "VL paradigm" . Additionally, some equations are lack sufficient explanation.
2. Table 1 shows significant performance fluctuations in the addition task (e.g., $V L_L^{TTC}$ drops to 51.40% at base 10), but no explanation is provided.
3. Although the experiments compare against DeepProblog, NeurASP, etc., the authors don't sufficiently discuss the limitations of these methods in unsupervised settings and why VL significantly outperforms them in specific tasks (e.g., 100% accuracy in addition), lacking in-depth qualitative analysis.


Strength:
1. The VL paradigm, by replacing reasoning with verification, eliminates the reliance on labels, filling a gap in unsupervised NeSy research.
2. The DCS algorithm reduces the complexity of COP from exponential to near-CSP levels, significantly improving computational efficiency. Time consumption data in Table 2 further validates its superiority.
3. The paper validates VL across four diverse tasks (addition, sorting, matching, chess), spanning simple arithmetic to complex rule-based scenarios. The results show excellent performance across tasks.

**Questions For Authors:**

See the strengths and weaknesses

**Relation To Broader Scientific Literature:**

enhance the trustworthy of model

**Theoretical Claims:**

N/A

---

> ### Author Rebuttal · Authors · 2025-03-31
>
> Dear reviewer NuWZ:
>
> Thank you for valuable comments.
>
> **Regarding Weakness 1:**
> About the language, you are absolutely correct. "KL" should be corrected to "VL"; this was a typographical error. Thank you for pointing it out.
> About the formulas, we have provided more detailed explanations for all of them. If anything remains unclear, please feel free to point it out:
> - **Equations (1) and (2)** summarize the training methods of previous label-based neuro-symbolic learning algorithms. Equation (1) generalizes the training method of algorithms like DeepProbLog, which utilize all candidate solutions for training. Equation (2) generalizes the training method of algorithms like ABL, which select the highest-scoring candidate solution from the search results for training.
> - **Equation (3)** describes the process of aligning the predicted probabilities from the neural network with the prior distribution of symbols, which helps prevent the network's predictions from collapsing into trivial solutions. For example, the addition rule is still satisfied even when the neural network predicts all symbols as 0.
> - **Equations (4) and (5)** describe the upper bound and mean error caused by the task itself in unsupervised neuro-symbolic systems. Equation (4) states that the error upper bound is equal to the sum of the distributions of all non-fixed-point symbols in the symbolic system, where a fixed point refers to symbols whose functions cannot be replaced in the system. Equation (5) states that the mean error is equal to the sum of the ratios between the distribution of each symbol and the number of its orbits, where the number of orbits represents how many other symbols can replace the function of a given symbol.
> - **Equation (6)** describes the empirical error caused by training a neural network in an unsupervised manner. Since the training error in unsupervised learning should be determined by its clustering error, the minimum error under all label assignments should be considered.
> - **Equation (7)** describes the upper-bound generalization error of an unsupervised neuro-symbolic system. This error is jointly determined by the training error, the complexity of the model space, and the task-specific error upper bound.
>
> **Regarding Weakness 2:**
> The observed performance variation is due to the early termination of the neural network training process before full convergence. To ensure an objective comparison, we maintained consistent hyperparameters across all methods and experimental settings. Specifically, we fixed the number of training epochs for all algorithms to 10 across different bases.
> In the particular setting you pointed out, while the accuracy was 51.40% at the 10th epoch, it exceeded 90% by the 13th epoch:
> |Epoch|10|11|12|13|14|15|
> |-|-|-|-|-|-|-|
> |$VL_{\not\perp}$|48.80|57.05|67.88|90.85|92.83|94.28|
> |$VL_{\not\perp}^{TTC} $ |51.40|63.05|78.13|97.38|98.85|98.95|
>
> This indicates that the algorithm did not fail; rather, due to the expanded symbolic space, the neural network requires more training. We did not adopt a higher number of epochs because this would cause DeepProbLog and NeurASP to exceed the time limit (300 hours).
>
> **Regarding Weakness 3:**
> In Section 3 of the original paper, we analyzed why methods such as DeepProbLog and NeurASP fail when reasoning without label Y, and why the VL paradigm remains effective even without relying on Y. Here, we further expand this analysis by incorporating specific tasks.
> The fundamental reason why DeepProbLog, NeurASP and etc. fail in unsupervised tasks is that they lack the ability to precisely identify the correct symbolic label S from a vast set of candidate symbolic labels candidates(S).
> - In supervised tasks, |candidates(S)| is relatively small, making this issue less severe.
> - In unsupervised tasks, |candidates(S)| is significantly larger, amplifying the problem and ultimately leading to algorithm failure.
>
> For example, in an **addition task**:
> - With supervision, the KB consists of addition rules. Given an equation like **Image1+Image2=Y**, when Y=0, we get candidates(S) = \{(0,0)\}; when Y=1, candidates(S) = \{(0,1), (1,0)\}, and so on. The average of |candidates(S)| is only **4.91**, indicating a relatively low level of imprecision.
> - Without supervision, where no Y is provided, given **Image1+Image2=Image3 Image4**, the KB alone must infer: candidates(S)=\{(0,0,0,0),...,(9,9,1,8)\}. Here, |candidates(S)| reaches **100**, making it infeasible for DeepProbLog and NeurASP to identify the correct symbolic label S efficiently. The overwhelming number of imprecise labels leads to training failure.
>
> In contrast, VL has a significant advantage in efficiently identifying the precise symbolic label S from candidates(S). VL excels even when |candidates(S)| is large because it leverages the proposed combinatorial ranking algorithm, which efficiently and precisely selects the optimal symbolic label S that adheres to the rules.

---

### Decision · Program_Chairs · 2025-05-01

**Decision:**

Accept (poster)

**Comment:**

he paper presents a Verification Learning (VL) that enables unsupervised neuro-symbolic (NeSy) systems to function effectively without reliance on labeled data. Traditional NeSy approaches depend heavily on labeled datasets, and removing labels often leads to challenges such as reduced symbolic information, an expanded solution space, and the emergence of shortcut solutions. VL addresses these issues by transforming the label-based reasoning process into a label-free verification process. In this framework, learning is guided solely by a verification function that checks whether the model's predictions adhere to predefined rules. Overall, the reviews are mixed, 2x weak reject and 2x weak accept, resulting in a borderline paper. The two negative reviews are rather short, and I have downweighted them. Still as pointed out in the reviews, the presentation has to be improved. More importantly, the paper should be discussed the work on learning by self-explaining, see e.g.

Wolfgang Stammer, Felix Friedrich, David Steinmann, Manuel Brack, Hikaru Shindo, Kristian Kersting. Learning by Self-Explaining. Transactions on Machine Learning Research (TMLR), 2024

as well as on the semtantic loss

Jingyi Xu, Zilu Zhang, Tal Friedman, Yitao Liang, Guy Van den Broeck: A Semantic Loss Function for Deep Learning with Symbolic Knowledge. ICML 2018: 5498-5507

and algorithmic supervision such as

Felix Petersen, Christian Borgelt, Hilde Kuehne, Oliver Deussen: Learning with Algorithmic Supervision via Continuous Relaxations. NeurIPS 2021: 16520-16531

Nevertheless, the idea of validation is interesting.